# Inotodiol induces hepatocellular carcinoma apoptosis by activation of MAPK/ERK pathway

Yushuang Xing[1,2]☯, Di Jia[3,4]☯, Xinping Zhu[1], Jialu Yang[3], Zhipeng Gao[3], Nana Meng[5], Haohao Xu[3], Mengxiao Wang[1], Shijun Chang[1], Mingqian Zhao[3], Shanbo Zhang[3], Zichen Mu[4], Qiang Tang[1,6]*, Weiming Zhao[1]*

1 Heilongjiang University of Traditional Chinese Medicine, Harbin, Heilongjiang, China, 2 Graduate Department, Qiqihar Medical University, Qiqihar, Heilongjiang, China, 3 Department of Medical Technology, Qiqihar Medical University, Qiqihar, Heilongjiang, China, 4 The Second Affiliated Hospital of Heilongjiang University of Chinese Medicine, Harbin, Heilongjiang, China, 5 Basic Medical Science College, Qiqihar Medical University, Qiqihar, Heilongjiang, China, 6 Rehabilitation Center, The Second Affiliated Hospital of Heilongjiang University of Chinese Medicine, Harbin, Heilongjiang, China

☯ These authors contributed equally to this work.
* zhaowm1969@126.com (WZ); tangqiang1963@163.com (QT)

**Data Availability Statement:** All relevant data are within the manuscript and its Supporting Information files.

**Funding:** This work was supported by the Heilongjiang Postdoctoral Fund (Grant

## Abstract

Hepatocellular carcinoma(HCC) has a high mortality and morbidity rate and seriously jeopardizes human life. Chemicals and chemotherapeutic agents have been experiencing problems such as side effects and drug resistance in the treatment of HCC, which cannot meet the needs of clinical treatment. Therefore, finding novel low-toxicity and high-efficiency anti-hepatocellular carcinoma drugs and exploring their mechanisms of action have become the current problems to be solved in the treatment of HCC. Several studies have reported anti-cancer effects of inotodiol. This study focuses on the anticancer effect of inotodiol in HCC cells and its molecular mechanism, aiming to explore its anticancer effect in depth. The CCK8 assay was utilized to assess cell viability, the scratch assay was utilized to detect migration ability, the clone formation assay was utilized to detect clonogenic ability, and flow cytometry was utilized to analyze apoptosis and cell cycle. Animal experiments was utilized to verify the inhibitory effect of inotodiol on HCC. Meanwhile, western blotting was utilized to detect proteins associated with apoptosis, cell cycle and MAPK/ERK pathway. These results showed that inotodiol has the ability to promote apoptosis, as well as inhibit the ability of cell proliferation, migration, and clonogenic ability. The cell cycle was arrested in G1 phase, when the expression of CDK2, CDK4, CDK6 and Cyclin D were inhibited. In addition, inotodiol showed to induce apoptosis, characterized by an increase in Bax expression, a decrease in Bcl-2, Bcl-XL and MCL1 expression, the initiation of cleaved PARP1 and cleaved caspase 3, and inhibition of the MAPK/ERK pathway. Animal studies demonstrated that inotodiol possessed the ability to suppress tumor growth in nude mice models, at the same time, there was no significant impact on the body weight and organs of the mice. In conclusion, the findings presented herein compellingly suggest that inotodiol may serve as a promising candidate for the treatment of hepatocellular carcinoma (HCC).

LBHZ23036), Heilongjiang University of Chinese Medicine funded projects (15041240014), 2021 Central Government's Plan to Support the Talent Training Project of the Reform and Development Fund of Local Universities, Heilongjiang Provincial Natural Science Foundation of China (Grant YQ2023H024). The funders had no role in study design, data collection and analysis, decision to publish, or preparation of the manuscript.

**Competing interests:** The authors have declared that no competing interests exist.

**Abbreviations:** 1.HCC, Hepatocellular carcinoma; 2.MAPK, Mitogen-activated protein kinase; 3.ERK, Extracellular regulated protein kinases; 4.MEK, Mitogen-activated protein kinase kinase; 5.RAF, Rapidly Accelerated Fibrosarcoma; 6.PARP1, Poly (ADP-ribose) polymerase 1; 7.BAX, BCL-2-associated X protein; 8.Bcl-2, B-cell CLL/lymphoma 2; 9.Bcl-XL, B-cell lymphoma-extra large; 10. MCL1, Myeloid cell leukemia 1; 11.RAS, Ras protein; 12.CDK2, Cyclin-dependent kinase 2; 13. CDK4, Cyclin-dependent kinase 4; 14.CDK6, Cyclin-dependent kinase 6; 15.DMEM, Dulbecco's Modified Eagle Medium; 16.FBS, Fetal Bovine Serum; 17.IC50, Half maximal inhibitory concentration; 18.PBS, Phosphate-buffered saline; 19.DMSO, Dimethyl sulfoxide; 20.PI, Propidium Iodide; 21.SDS, Sodium Dodecyl Sulfate; 22.PVDF, Polyvinylidene fluoride; 23.ECL, Enhanced chemiluminescence; 24.GAPDH, Glyceraldehyde 3-phosphate dehydrogenase; 25.SEM, Standard error of the mean; 26.CDKs, Cyclin-dependent kinases; 27.p38 MAPK, p38 mitogen-activated protein kinase; 28.JNK, c-Jun NH2-terminal kinase.

## Introduction

HCC represents a form of primary liver cancer, characterized by its significant heterogeneity, propensity for metastatic spread, and generally unfavorable outcomes, with a steadily increasing incidence rate and a serious threat to people's life and health [1–3]. In recent years, more and more anti-hepatocellular carcinoma drugs have been successfully marketed and applied in the clinic. Although the clinical first-line anti-hepatocellular carcinoma drugs sorafenib, lenvatinib, donafenib and chemotherapeutic drugs cisplatin and adriamycin can slightly prolong the life time of the patients, they have the problems of low drug efficacy, short drug-resistance cycle and poor prognosis, and the common adverse reactions mainly include hypertension, hemorrhage, and neuropathy and so on [4]. Therefore, actively searching for new anti-hepatocellular carcinoma drugs with good efficacy, low toxicity and side effects, and low drug resistance is a problem we urgently need to solve.

Inonotus obliquus is a kind of medicinal and edible fungi parasitized on living birch trees, which can absorb the nutrients of birch trees, and is often taken in water for the prevention and treatment of diabetes mellitus, gastrointestinal diseases and tumors, which is mild and has no side effects. In the 1950s, the Medical College of Russia reported a number of clinical trials applying the preparations of inonotus obliquus for the treatment of malignant tumors such as gastric, esophageal and breast cancers, and found that inonotus obliquus has the characteristics of inhibiting the reproduction and prolonging the survival, with little side effects [5]. Inotodiol is a lanolinane-type triterpenoid active compound unique to inonotus obliquus, which has a variety of biological activities and can widely inhibit the growth and proliferation of a variety of tumors, and has no significant adverse effects when used in vivo. It has been found that inotodiol can inhibit human ovarian cancer transplantation tumor in nude mice [6], and induce apoptosis in human cervical cancer HeLa cells [7], human ovarian cancer SKOV3 cells [8], and human lung cancer A549 cells [9], but the mechanism of apoptosis induction has not been explored in depth. Currently, only one study has reported the anti-hepatocellular carcinoma effect of inotodiol, which demonstrated that inotodiol is capable of markedly suppressing the growth of HCC cells, and the target of its action was HIF-1α [10]. The article only drew a conclusion through PCR detection of HIF-1α gene expression, and failed to investigate its mechanism of action, which limited the development and clinical application of inotodiol. This study will explore the specific mechanism of action of inotodiol in the treatment of HCC, with a view to providing theoretical basis and data support for the advancement and utilization of future pharmaceuticals.

Activation of mitogen-activated protein kinase(MAPK) /extracellular regulated protein kinases(ERK) pathway has been associated with the progression of several malignancies. The function of MAPK/ERK pathway in HCC has attracted much attention in recent years. Several studies found that natural plant extracts can exert anti-hepatocellular carcinoma effects by inhibiting the MAPK/ERK pathway. Flavonoids [11], rutaecarpine glycosides [12], and anisomycin [13] inhibit the proliferation and promote apoptosis of HCC cells by harmonizing the MAPK pathway. In our research, we have identified for the first time that inotodiol triggers apoptosis in HCC cells and inhibited the MAPK/ERK pathway at the same time. This study identifies a new pathway of inotodiol against HCC, and also provides a new idea for future clinical application of inotodiol in the treatment of HCC.

## Materials and methods

### Chemicals and antibodies

Inotodiol(CAS: 35963-37-2) was purchased from Wuhan Tianshi Biotechnology, cyclophosphamide was purchased from Shanghai yuanye Bio-Technology Co., Ltd(CAS: S30563), CCK8

kit, 10×Tris-Glycine-SDS electrophoresis Buffer, 10×TBST, 10×Electrophoresis Transfer Buffer, Rapid Closure Solution and Easy PAGE Color Rapid Gel Preparation Kit were purchased from Severn Innovation, RIPA lysate and PMSF (100 mM) was purchased from Biyuntian Biotechnology, DMSO was purchased from BioFroxx, AnnexinV-FITC Apoptosis Detection Kit was purchased from Bioss, Color Prestained Protein Marker was purchased from Yamei and Bioswamp. Primary antibodies such as cleaved caspase 3(ab32042), cleaved PARP1(ab32064) were purchased from abcam, BAX(#AF0120), Bcl-2(#AF6139), Bcl-XL (#AF6414), MCL1(AF5311), GAPDH(#AF7021), Tublin(#AF7011), RAS(#AF0247), c-RAF (#AF6065), p-c-RAF(#AF3065), ERK1/2(#AF0155), p-ERK1/2(#AF1015), MEK1/2(#AF6385), p-MEK1/2(#AF8035), CDK2(#AF6237), CDK4(#AF4034), CDK6(#DF6448), cyclin D (#AF0931) were purchased from Affinity Biosciences, Goat Anti-Rabbit IgG (H+L) HRP secondary antibody(#S0001) and Goat Anti-Mouse IgG (H+L) HRP secondary antibody(#S0002) were purchased from Affinity Biosciences.

## Cell culture

In this investigation, HepG2, and HCCLM3 human HCC cell lines were purchased from Otwo Biotech and cultivated using Dulbecco's Modified Eagle Medium (DMEM) (Gibco, USA) enriched with 10% fetal bovine serum (FBS) (BI, Israeli), and 1% penicillin-streptomycin solution. Additionally, the sk-hep-1 human HCC cell line was grown in RPMI 1640 medium (Gibco, USA), also supplemented with 10% FBS and 1% penicillin-streptomycin solution. Human liver cells THLE-2 and mouse liver cells AML12 were respectively cultured in their specific culture media (Pricella, China). Cultivation for all cell lines was conducted in a $CO_2$ incubator (Thermo Scientific, USA), under conditions of 5% $CO_2$ and a temperature of 37°C.

**Animal husbandry.** Male BALB/c nude mice, 4 weeks old, with a body weight of 18 to 22g, were provided by Beijing Vital River Laboratory Animal Technology Co., Ltd., with an animal license number SCXK (Beijing) 2021–0006. The animals were housed in an environment with a temperature of 24 to 25°C, relative humidity of 50% to 60%, and a 12-hour light-dark cycle, with free access to water and food during the breeding period.

## CCK8

HCC cells were seeded into 96-well microplates at a concentration of $5×10^3$ cells per well and then maintained for 24 hours at a temperature of 37°C within an atmosphere-controlled chamber that included 5% $CO_2$. Following this, after a further 48 or 72 hours of culture, the CCK8 reagent was introduced into each well for a suitable duration to facilitate the reaction. The median inhibitory concentration ($IC_{50}$) for cell proliferation was calculated employing the Logit model, utilizing the analytical capabilities of GraphPad Prism version 9.4.

## Wound healing assay

The cells were cultivated in six-well culture plates until they achieved a confluence of 80–90%. Subsequently, a wound healing assay was conducted. The surface of the plate was uniformly scored in a straight line using a 10 μL sterile pipette tip, followed by 1–2 washes with phosphate-buffered saline(PBS) to eliminate any detached cells. Thereafter, the cells were incubated in a serum-deprived DMEM or 1640 medium enriched with inotodiol at a concentration equivalent to their $IC_{50}$. Images were captured at intervals of 0, 12, 24, and 48 hours to assess the area of cell migration and to determine the migration efficiency. The migration efficiency was determined by applying the following formula: Migration rate = [(Initial wound area—Final wound area) / Initial wound area] × 100%[13].

## Clone formation assay

HepG2 and sk-hep-1 cells were seeded in six-well culture plates at an initial density of 500 cells per well. Upon achieving cell adherence, the cultures were exposed to either a medium supplemented with dimethyl sulfoxide (DMSO) or one containing inotodiol for a period of 48 hours. Subsequently, the cells were transferred to a medium devoid of any drug substances and maintained for a duration of two weeks. For histological examination, the cells were immobilized using a 4% paraformaldehyde solution for 15 minutes, followed by staining with crystal violet for a period of 20 minutes, after which they were documented photographically.

## Apoptosis rate assay

Apoptosis was assessed through flow cytometric evaluation utilizing the Annexin V-FITC and propidium iodide (PI) staining protocol. Following an initial 24-hour culture period, the cells underwent treatment with inotodiol for a subsequent 48-hour interval. HepG2 and sk-hep-1 cells were harvested, rinsed with chilled PBS, and resuspended in a solution containing 5 μL of Annexin V along with 5 μL of PI. The cell suspension was carefully agitated, then incubated in darkness for a duration of 20 minutes, followed by a washing step. The samples were subsequently subjected to flow cytometric analysis.

## Cell cycle assay

HepG2 and sk-hep-1 cells were exposed to inotodiol for a period of 48 hours. Subsequently, the cells underwent a washing process with chilled PBS and were then immersed in a bath of cold 70% ethanol for an extended period. After this, the cells were once again washed with chilled PBS and subsequently suspended in a PI solution, followed by incubation at a temperature of 37°C for 30 minutes. The cells were then ready for flow cytometric analysis, the cell cycle distribution was determined using the ModFit LT 3.1 software.

## Western blotting analysis

Proteins from the cells were resolved on a sodium dodecyl sulfate (SDS) polyacrylamide gel through electrophoresis and subsequently transferred onto a polyvinylidene fluoride (PVDF) membrane. The membranes were pre-equilibrated with a Rapid Block Solution for a period of 15 minutes at ambient temperature, followed by an extended incubation period with the primary antibody at 4°C overnight. Subsequently, the membranes were exposed to a secondary antibody for a period of 2 hours at room temperature while agitating on a platform shaker. The presence of proteins was revealed by employing an enhanced chemiluminescence (ECL) detection system, as per the manufacturer's protocol. Glyceraldehyde 3-phosphate dehydrogenase (GAPDH) and tubulin served as endogenous control proteins for normalization purposes.

**Animal experiments.** BALB/c nude mice were randomly divided into four groups: the control group (physiological saline), the positive drug group (cyclophosphamide, 20mg/kg/d), the high-dose inotodiol group (inotodiol, 20mg/kg/d), and the low-dose inotodiol group (inotodiol, 5mg/kg/d), with three mice in each group. HepG2 cells were suspended in serum-free cell culture medium to a concentration of $2\times10^7$cells/mL, and 100μL of the cell suspension was injected subcutaneously into the right axillary region of the nude mice to establish a subcutaneous tumor-bearing mouse model. After the subcutaneous tumors were formed, drug administration was initiated. Two weeks after drug administration, the mice were weighed, subcutaneous tumors and various organs were excised, photographed, weighed, and the volume of the subcutaneous tumors was calculated.

### Statistical analyses

Statistical analysis of the data was conducted using the t-test within the GraphPad Prism version 9.4 software, with results presented as the mean value accompanied by the standard error of the mean (SEM). Each experimental trial was conducted autonomously on no fewer than three occasions, with each condition being replicated thrice. A p-value of less than 0.05 was established as the threshold for deeming a result to be statistically significant.

## Results

### Inotodiol inhibit cell viability of HCC cells

To ascertain the influence of inotodiol on the growth potential of HCC cells, an initial assessment of cellular viability was performed using the CCK8 assay. The cell lines sk-hep-1, HepG2 and HCCLM3 were subjected to various doses of inotodiol and cultivated in 1640 or DMEM media with inotodiol for 48 hours (Fig 1A) and 72 hours (Fig 1B). The findings indicated that inotodiol exerted an suppressive effect on cell proliferation in a dose-dependent fashion, with the sk-hep-1 and HepG2 cell lines exhibiting the highest sensitivity to the compound. The calculated $IC_{50}$ values for inotodiol after a 48-hour exposure for sk-hep-1 and HepG2 cells were 55.48±1.94 μg/mL and 77.26±1.12 μg/mL, respectively. Subsequent experiments were conducted by exposing sk-hep-1 and HepG2 cells to inotodiol at the determined $IC_{50}$ concentration for a period of 48 hours.

To assess the safety profile of inotodiol, we conducted experiments in which THLE-2 and AML12 cells were exposed to a range of inotodiol concentrations. The results shown in Fig 1A and 1B indicate that the IC50 values for both THLE-2 and AML12 cells after 48 and 72 hours of inotodiol treatment were significantly higher compared to those observed in HCC cells. The calculated IC50 values of inotodiol after a 48-hour exposure for THLE-2 and AML12 cells were 476.73±3.21 μg/mL and 427.57±5.16 μg/mL, respectively. This suggests that inotodiol exhibits a reducedcytotoxic effect relative to its impact on HCC cells.

### Inotodiol inhibits the migration of HCC cells

Administration of inotodiol at the $IC_{50}$ concentration significantly hindered the migratory capacity of sk-hep-1 and HepG2 cells, as illustrated in Fig 2. The inhibitory effect of inotodiol

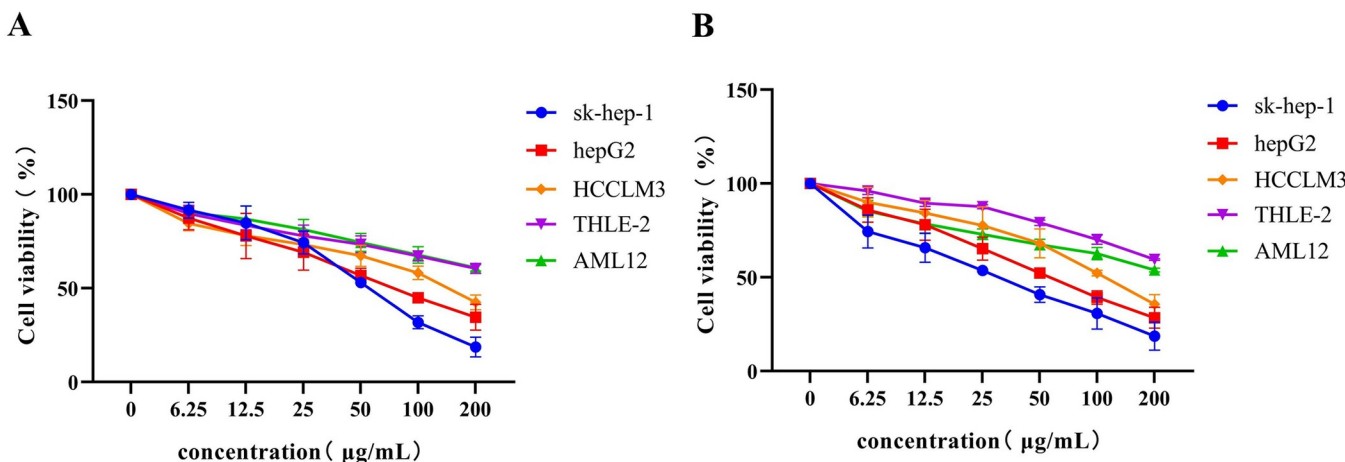

**Fig 1. Inotodiol inhibits HCC cell viability.** (A)The influence of inotodiol on the survival of HCC cells and liver cells was examined over a 48-hour timeframe using the CCK-8 assay with a range of concentrations, n = 3. (B)The influence of inotodiol on the survival of HCC cells and liver cells was examined over a 72-hour timeframe using the CCK-8 assay with a range of concentrations, n = 3.

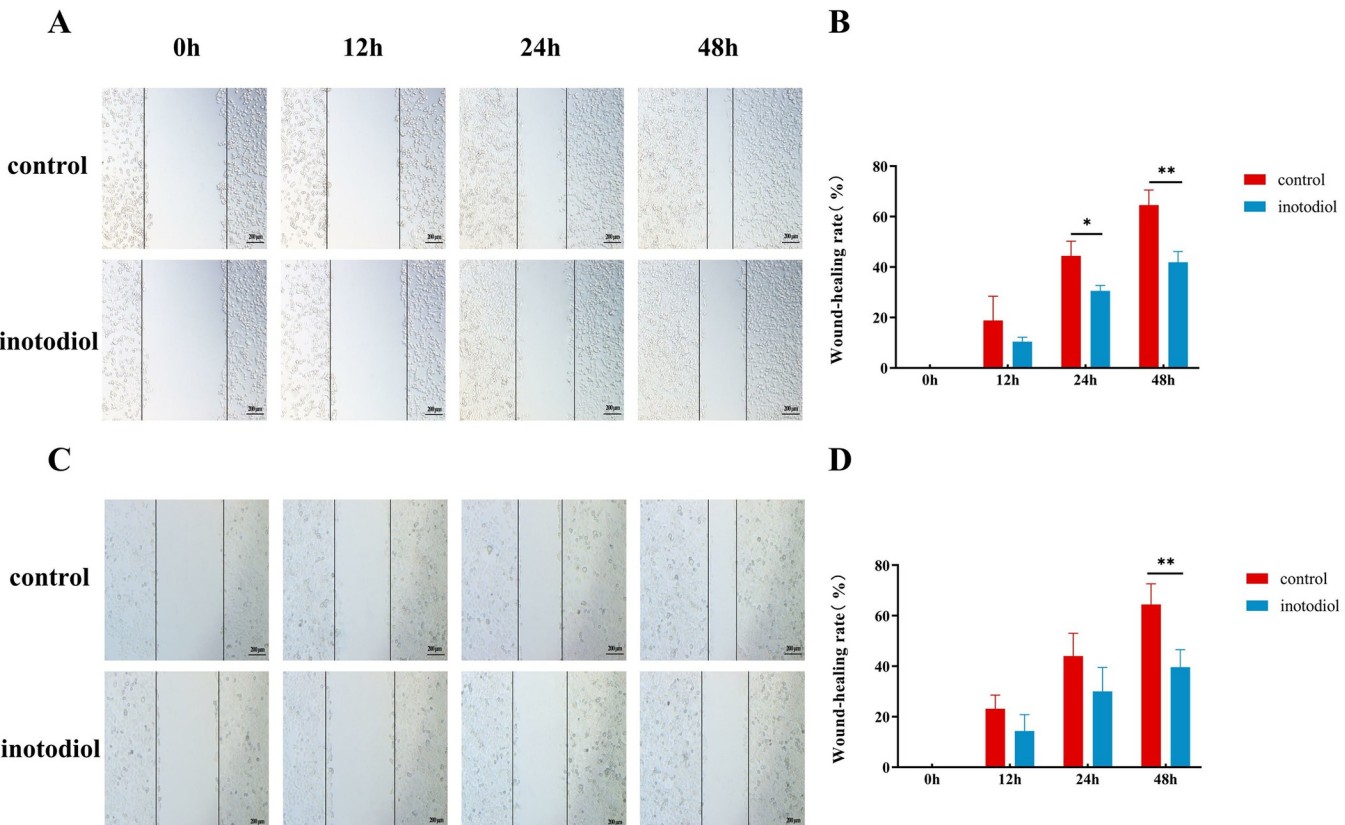

**Fig 2. Inotodiol inhibits sk-hep-1, hepG2 cells migration.** (A-B)Effect of inotodiol treatment of sk-hep-1 cells on the healing area of scratches and statistical results. (C-D)Effect of inotodiol treatment of hepG2 cells on the healing area of scratches and statistical results. *significantly different from control group. *p<0.05, **p<0.01, n = 3.

on the movement of these cell lines was notably pronounced after a 48-hour exposure period when contrasted with untreated control groups.

## Inotodiol decreases clone formation in HCC cells

To delve deeper into the suppressive impact of inotodiol on the proliferation of HCC cells, subsequent colony formation assays were conducted. The findings distinctly illustrated a reduction in the quantity of colonies for both the neoplastic cell lines subsequent to treatment with inotodiol (Fig 3).

## Inotodiol induces cell cycle arrest in HCC cells

Further investigation into the impact of inotodiol on HCC cells was conducted through flow cytometry to ascertain if the compound's inhibitory impact on growth is mediated by cell cycle arrest. The findings indicated a notable increase in the population of sk-hep-1 and HepG2 cells in the G1 phase when exposed to inotodiol (Fig 4A–4D). Concurrently, western blot analysis substantiated that inotodiol induced a halt in the G1 phase of the cell cycle for these cell lines, attributable to diminished levels of expression of proteins pivotal to the G1-S phase progression, including CDK2, CDK4, CDK6 and cyclin D (Fig 4E–4H).

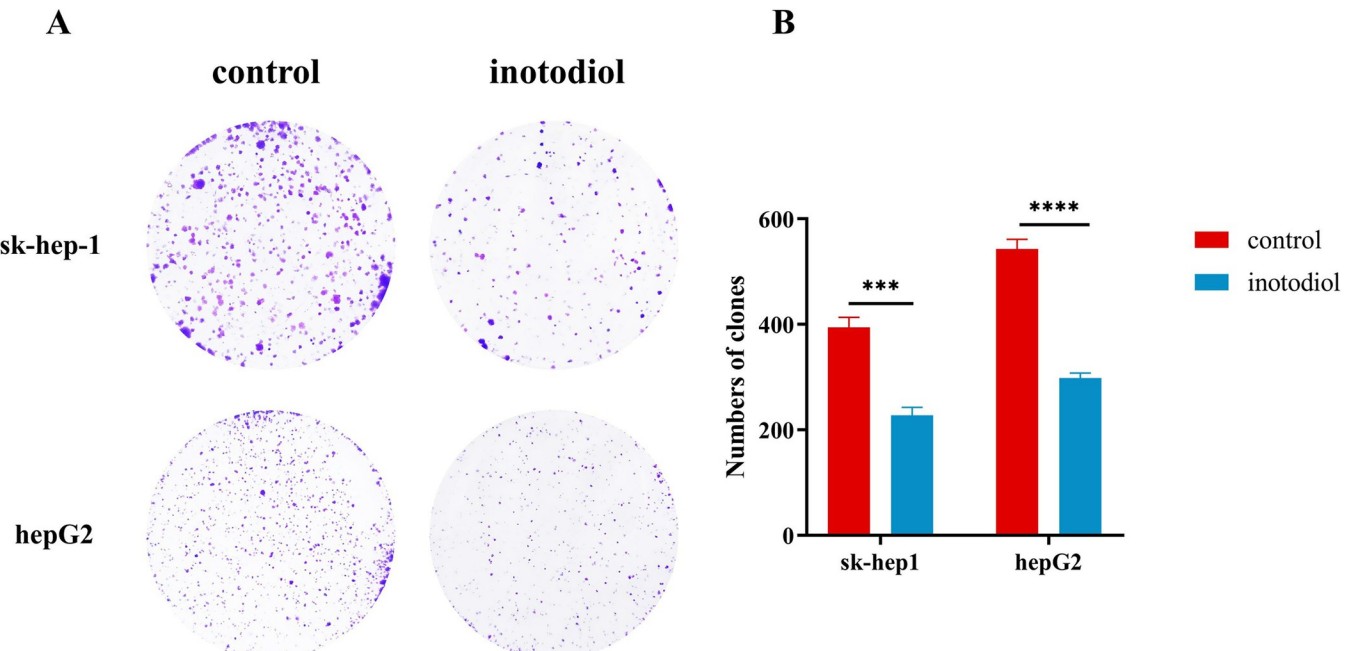

**Fig 3. Inotodiol decreases clone formation in HCC cells.** (A-B) The Clone formation assay and the number of clones. *significantly different from control group. ***p<0.001, ****p<0.0001, n = 3.

## Inotodiol promotes HCC cells apoptosis

We next detected apoptosis using Annexin V-FITC and PI assay kits and analyzed them via flow cytometry to further explore the possible molecular mechanisms that inhibit HCC cell viability. The findings revealed that inotodiol triggered substantial apoptosis, with apoptosis rates showed (71.30±1.18)% and (52.70±0.98)% in sk-hep-1 and hepG2 cells, respectively (Fig 5A and 5B). The next western blot assay confirmed that inotodiol provoked apoptotic effects in HCC cells. Our findings indicated that inotodiol markedly elevated the levels of the pro-apoptotic protein Bax and concurrently reduced the levels of the anti-apoptotic protein Bcl-2, Bcl-XL, MCL1. These alterations in protein expression are pivotal in modulating the apoptotic pathway. In addition, following treatment with inotodiol, an increase in the levels of cleaved-PARP1 and cleaved-caspase 3 proteins was observed in both the sk-hep-1 cells (Fig 5C and 5D) and hepG2 cells (Fig 5E and 5F). These results clearly indicate that inotodiol promotes apoptosis in HCC cells.

## Inotodiol inhibits Ras-Raf-MEK-ERK pathways in HCC cells

A multitude of investigations have correlated the initiation of apoptosis with the activity of the MAPK signaling cascade. With the aim of assessing the influence of inotodiol on this pathway, we performed an analysis of the protein expression within the MAPK signaling cascade in HCC cells utilizing western blotting. As depicted in Fig 6, the overall expression levels of c-RAF, ERK1/2, and MEK1/2 proteins remained unchanged in response to inotodiol treatment. In contrast, a decrease was observed in the phosphorylation levels of c-RAF, ERK1/2, MEK1/2, and RAS. These findings validate that inotodiol modulates the Ras-Raf-MEK-ERK signaling pathways in HCC cells.

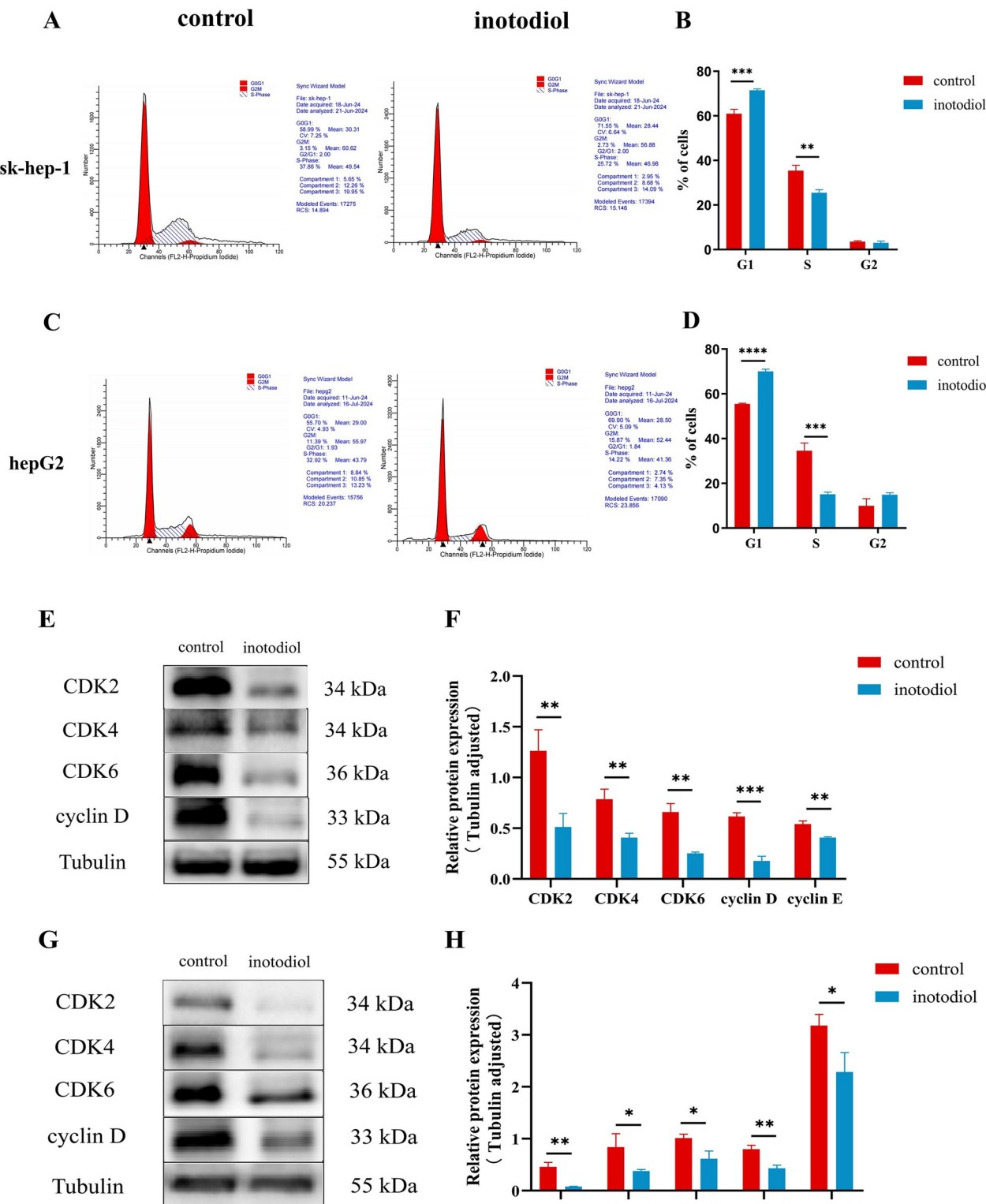

**Fig 4. Inotodiol-mediated cell cycle halt at the G1 phase in HCC cells.** (A-D) Inotodiol elevated the prevalence of cells within the G1 phase of the cell cycle. (E-F) Western blotting assay for the analysis of CDK2, CDK4, CDK6, cyclin D expression in inotodiol-treated sk-hep-1 cells at indicated concentration for 48h. (G-H) Western blotting assay for the analysis of CDK2, CDK4, CDK6, cyclin D expression in inotodiol-treated hepG2 cells at indicated concentration for 48h. *significantly different from control group. *p<0.05, **p<0.01, ***p<0.001, ****p<0.0001, n = 3.

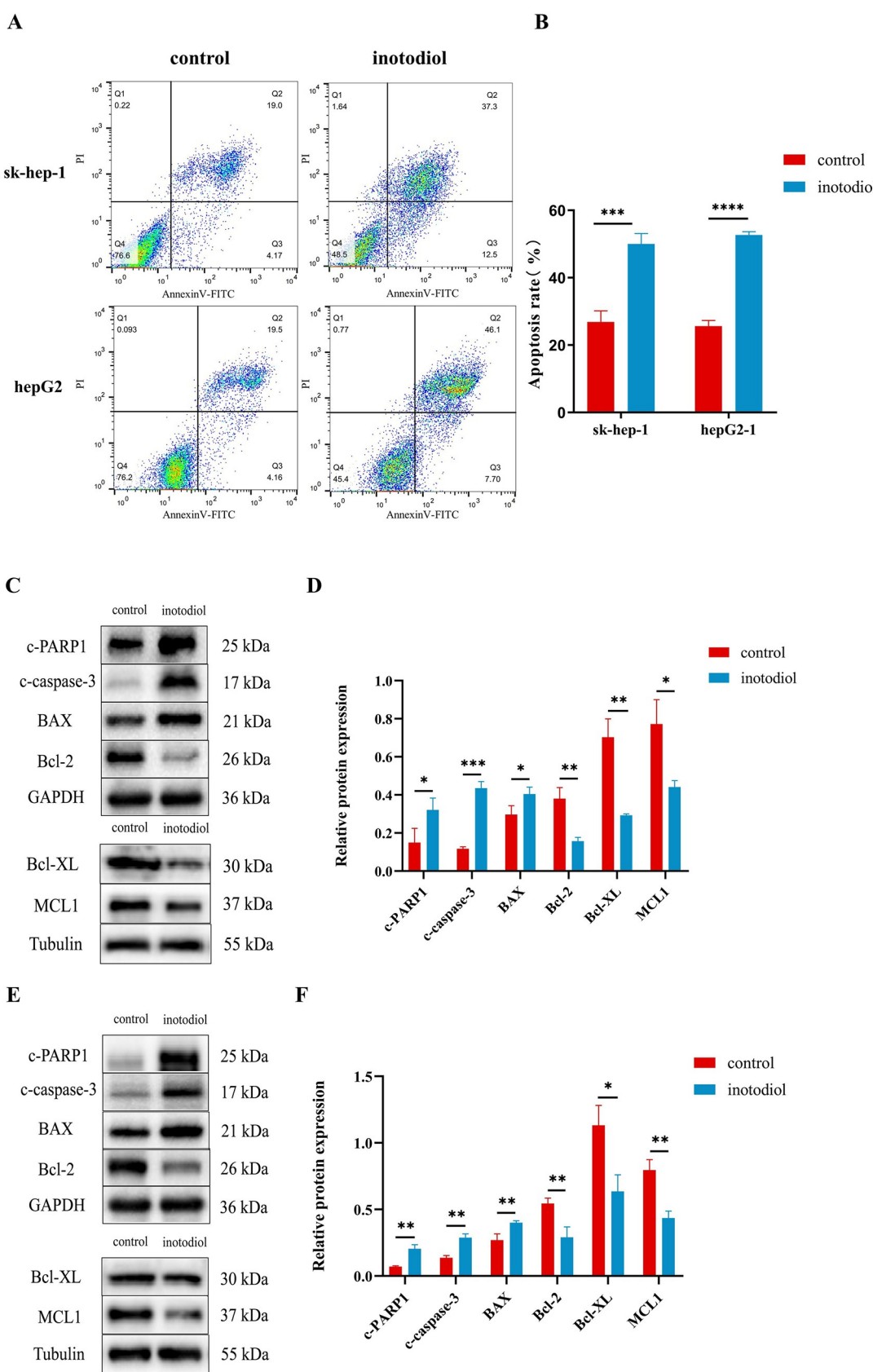

**Fig 5. Inotodiol triggered apoptosis in HCC cells.** (A-B) The apoptosis rate of inotodiol-treated sk-hep-1 and hepG2 cells after 48 hours was detected using flow cytometry. (C-F) Examination of the expression profiles of proteins associated with apoptosis in the sk-hep-1 (C-D) and hepG2 (E-F) cell lines was conducted using western blotting after the treatment with inotodiol.*significantly different from control group. *$p<0.05$, **$p<0.01$, ***$p<0.001$, ****$p<0.0001$, n = 3.

## Inotodiol inhibits tumor growth in mice without affecting organ function

In comparison to the control group, the tumor weight and volume were found to be diminished in the inotodiol high-dose group, inotodiol low-dose group, and the positive drug group, with the most pronounced reduction occurring in the inotodiol high-dose group (Fig 7, Table 1). After the administration of inotodiol, the body weight and the weight of various organs in the mice showed no significant changes (Table 2).

## Discussion

HCC is not easy to be diagnosed in its initial stages, and HCC cells are not sensitive to radiotherapy, so it is often treated by surgery, but it is very easy to recur after surgery, resulting in a five-year survival rate that remains around a mere 10%. Therefore, exploring the mechanism of HCC and searching for effective therapeutic drugs hold substantial importance for the clinical management of HCC [14]. Inotodiol is one of the anti-tumor active ingredients in the seed bodies of Inonotus obliquus, and several studies have reported its antitumor activity. It was found that inotodiol suppressed cell growth and triggered apoptosis of Hela cells [7], breast cancer MCF-7 cells [15,16], and human ovarian cancer SKOV3 cells [17]. In this study, it was established that inotodiol markedly inhibited the activity of HCC cells in a concentration-dependent manner by CCK8 assay. At the same time, we found that the IC50 value of inotodiol in normal liver cells is much greater than in HCC cells, indicating that our drug concentration causes less damage to normal liver cells. Previous studies have found that after administering inotodiol to mice for 13 weeks, there were no deaths or signs of organ abnormalities in the mice. Additionally, it did not affect the mice's blood, biochemical indicators and cytokines, indicating that inotodiol exhibits non-toxic or very low toxicity in vivo and is relatively safe to use [18–20]. In vivo experiments have demonstrated that inotodiol possesses the ability to suppress tumor growth in mice, at the same time, there were no deaths and no obvious toxic reactions in the mice, and there were no significant changes in body weight and organ mass. This once again confirms the low toxicity of inotodiol. Flow cytometry assay revealed that HCC cell apoptosis increased significantly after inotodiol treatment. Previous investigations have found that inotodiol can trigger apoptosis in HCC cells through the suppression the Bax/Bcl-2 pathway, and the present study delved into the mechanism of apoptosis induced by inotodiol. Western blot detection of apoptosis-related proteins revealed that the expression of cleaved Caspase-3, cleaved PARP1, Bax proteins decreased and the expression of Bcl-2, Bcl-XL, MCL1 proteins increased by inotodiol treatment. This indicates that the induction of apoptosis by inotodiol is multi-targeted, which provides a reference for the clinical study of inotodiol in the later stage, and also provides data support for solving the problem of drug resistance in clinical use. Research has found that the camptothecin analog FL118 can induce p53/p21-dependent senescence and also induce p53-independent apoptosis, both of which contribute to the inhibition of colon cancer cell growth [21]. The late-stage autophagy inhibitor bafilomycin A1 promotes apoptosis [22], and after knocking down the key autophagy protein Beclin1, apoptosis is significantly reduced [23]. Previous studies have confirmed that apoptosis is intricately linked with senescence and autophagy. Our preliminary study explored the effects of inotodiol on autophagy and senescence in HCC cells, and the results show that inotodiol may induce autophagy and senescence in HCC cells (S1 and S2 Figs). While the intricate dynamics between

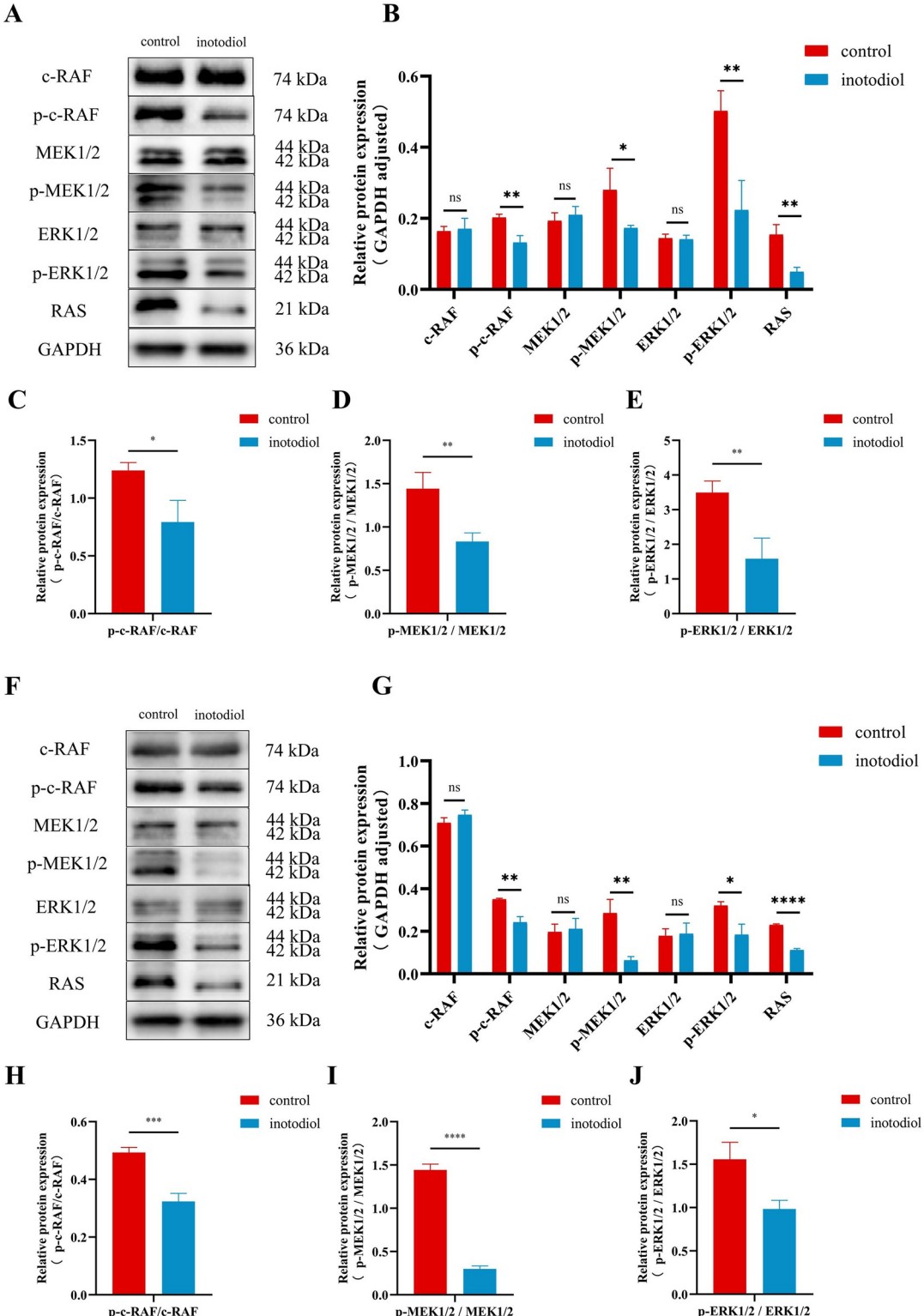

**Fig 6. Inotodiol inhibits MAPK/ERK pathways in HCC cells.** (A-E) Protein expression and statistical results of RAS/RAF/MEK/ERK pathway related proteins in sk-hep-1 cells ascertained by Western Blot. (F-J) Protein expression and statistical results of MAPK/ERK pathway related proteins in hepG2 cells detected by Western Blot. NS, not significant different from control group, *significantly different from control group. *p<0.05, **p<0.01, ***p<0.001, ****p<0.0001, n = 3.

**A**                                         **B**

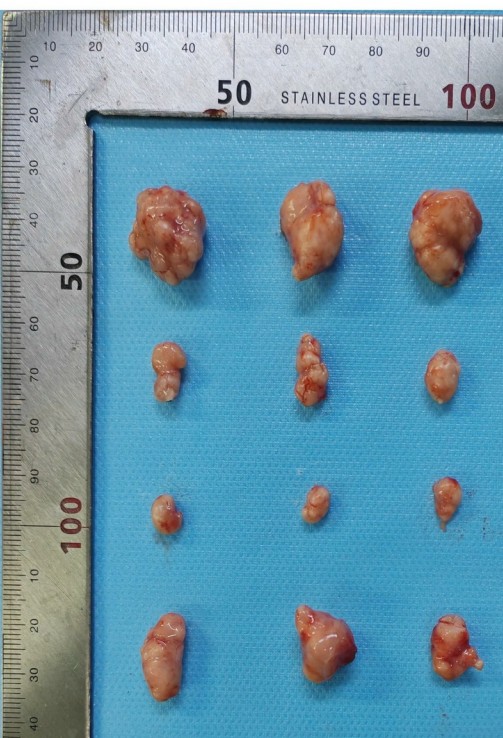

**Fig 7. Inotodiol inhibits the growth of liver cancer tumors in mice.** (A-B) Photos of subcutaneous tumors in mice from each group. *significantly different from control group. *p<0.05, ***p<0.001, ****p<0.0001, n = 3.

autophagy, senescence, and apoptosis remain to be fully elucidated, our forthcoming research will proactively delve into these interactions.

Disruption of cell cycle regulation is a critical element contributing to the uncontrolled expansion of cancerous growths. The deregulation of two pivotal cell cycle checkpoints, namely the G1/S transition and the G2/M transition, can directly lead to the abnormal proliferation of tumor cells [24]. The progression through the cell cycle is predominantly governed by the interplay of cyclin-dependent kinases (CDKs), cyclins, and intrinsic CDK inhibitors. Among them, CDK is the main cell cycle regulator, which can bind to cyclin to form cyclin-CDK complex and regulate interphase and mitotic progression [25,26], CDKs involved in the

**Table 1. The effect of inotodiol on the tumor volume and weight in tumor-bearing mice.**

| Group | Tumor volume($mm^3$) | Tumor weight(g) |
|---|---|---|
| Control group | 2119.73±106.71 | 1.54±0.18 |
| Positive drug group | 332.27±28.2**** | 0.3±0.03*** |
| Inotodiol high-dose group | 136.85±45.3**** | 0.13±0.03*** |
| Inotodiol low-dose group | 826.81±195*** | 0.82±0.31* |

*significantly different from control group.

*p<0.05

***p<0.001

****p<0.0001, n = 3.

**Table 2. The effect of inotodiol on the body weight and organ mass of tumor-bearing mice.**

| Group | Body weight(g) | Heart(g) | Liver(g) | Spleen(g) | Lung(g) | Kidney(g) |
|---|---|---|---|---|---|---|
| Control group | 30.93±0.49 | 0.17±0.04 | 1.72±0.08 | 0.28±0.03 | 0.22±0.09 | 0.5±0.07 |
| Positive drug group | 31.63±2.4 | 0.12±0.01 | 1.72±0.22 | 0.2±0.02 | 0.2±0.01 | 0.55±0.07 |
| Inotodiol high-dose group | 30.47±1.51 | 0.23±0.02 | 1.59±0.14 | 0.2±0.03 | 0.19±0.04 | 0.56±0.02 |
| Inotodiol low-dose group | 32.47±1.55 | 0.16±0.03 | 1.74±0.09 | 0.26±0.06 | 0.22±0.01 | 0.59±0.04 |

regulation of G1 phase include CDK2, CDK4 and CDK6, and cyclins include cyclin D and cyclin E [27,28]. Some studies have found that targeting CDK4/CDK6 can treat gastric cancer, and the main mechanism by which this approach exerts its effects is by impeding the cell cycle's progression from the G1 phase to the S phase, thereby curbing the expansion and proliferation of malignant cells [29]. It has been found that inonotus obliquus triggers apoptosis in several types of cancer cells, including colorectal cancer HCT-116 cells [30], HCC hepG2 cells [31], esophageal cancer EC-109, EC-9706 cells [32]. Additionally, it has been observed to cause an halt in the G1 phase of cellular proliferation. However, to date, no studies have specifically examined the impact of inotodiol on tumor cell cycle regulation. In this study, we found that HCC cells stayed in G1 phase increased after inotodiol treatment by flow cytometry assay, and western blot experiments found that inotodiol decreased CDK2, CDK4, CDK6, Cyclin D protein expression. These findings indicate that inotodiol suppresses cellular multiplication through the initiation of cell cycle halt in the G1 phase.

MAPK pathways, including ERK pathway, p38 mitogen-activated protein kinase (p38 MAPK), and c-Jun NH2-terminal kinase (JNK), are involved in varieties of physiopathological procedures, including cell proliferation, differentiation, and apoptosis [33], and MAPK/ERK pathway, as a major pathway, is associated with the uncontrolled growth characteristic of numerous cancerous cells [34,35]. Research revealed that excessive stimulation of the MAPK/ERK pathway leads to the transformation of regular hepatocytes into a malignant state and enhances the proliferation, migration, and invasive capabilities of HCC cells [36–38]. We utilized western blot analysis to identify the pivotal proteins within the MAPK/ERK cascade, including RAS, RAF, MEK, and ERK. The findings indicated that the comparative levels of RAS, p-c-RAF, p-ERK, and p-MEK proteins were reduced by inotodiol treatment, suggesting that inotodiol could inhibit the MAPK/ERK pathway.

To conclude, inotodiol has been demonstrated to inhibit the proliferation of HCC cells and induce apoptosis, with in vivo studies confirming its anti-HCC effects. This study marks the first instance of uncovering the potential mechanism by which inotodiol exerts its anti-HCC action, which may involve the modulation of the MAPK/ERK signaling pathway. This study by targeting this pathway, inotodiol could lead to cell cycle arrest at the G1 phase, thereby inhibiting the progression of liver cancer. This mechanism is significant as it suggests a specific molecular target for inotodiol's therapeutic effects, potentially offering a new avenue for HCC treatment. The accomplishment of this research is poised to open up innovative pathways for the progression of novel, efficient and safe drugs for clinical liver cancer treatment, which will help to improve the current situation of clinical liver cancer medication, and has important theoretical significance and practical value.

## Supporting information

**S1 Fig. The effect of inotodiol on HCC cell senescence.** *significantly different from control group. ***$p<0.001$, ****$p<0.0001$, n = 3.
(TIF)

**S2 Fig. The effect of inotodiol on autophagy in HCC cells.** *significantly different from control group. ***p<0.001, ****p<0.0001, n = 3.
(TIF)

**S3 Fig. Inotodiol inhibits HCC cells migration.**
(TIF)

**S4 Fig. Inotodiol decreases clone formation in HCC cells.**
(TIF)

**S5 Fig. Inotodiol-mediated cell cycle halt at the G1 phase in sk-hep-1 cells.**
(TIF)

**S6 Fig. Inotodiol-mediated cell cycle halt at the G1 phase in hepG2 cells.**
(TIF)

**S7 Fig. Inotodiol triggered apoptosis in HCC cells.**
(TIF)

**S1 Table. The effect of inotodiol on sk-hep-1 cell viability after 48 or 72 hours of treatment.**
(TIF)

**S2 Table. The effect of inotodiol on hepG2 cell viability after 48 or 72 hours of treatment.**
(TIF)

**S3 Table. The effect of inotodiol on HCCLM3 cell viability after 48 or 72 hours of treatment.**
(TIF)

**S4 Table. The effect of inotodiol on THLE-2 cell viability after 48 or 72 hours of treatment.**
(TIF)

**S5 Table. The effect of inotodiol on AML12 cell viability after 48 or 72 hours of treatment.**
(TIF)

**S6 Table. The effect of inotodiol on the tumor volume and weight in tumor-bearing mice.**
(TIF)

**S7 Table. The effect of inotodiol on the body weight and organ mass of tumor-bearing mice.**
(TIF)

**S1 Raw images. The original blot and gel images contained in the manuscript's main figures.**
(PDF)

## Author Contributions

**Conceptualization:** Mengxiao Wang.

**Data curation:** Zhipeng Gao, Haohao Xu, Mingqian Zhao.

**Formal analysis:** Zichen Mu.

**Funding acquisition:** Weiming Zhao.

**Project administration:** Di Jia, Qiang Tang, Weiming Zhao.

**Resources:** Jialu Yang, Mengxiao Wang.

**Software:** Shijun Chang, Shanbo Zhang.

**Validation:** Xinping Zhu, Nana Meng.

**Writing – original draft:** Yushuang Xing.

**Writing – review & editing:** Yushuang Xing.

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
