## [Decision Letter · Decision Letter 0]

10 Oct 2024

PONE-D-24-40336Inotodiol induces hepatocellular carcinoma apoptosis by activation of MAPK/ERK pathwayPLOS ONE

Dear Dr. Xing,

Thank you for submitting your manuscript to PLOS ONE. After careful consideration, we feel that it has merit but does not fully meet PLOS ONE’s publication criteria as it currently stands. Therefore, we invite you to submit a revised version of the manuscript that addresses the points raised during the review process.

We look forward to receiving your revised manuscript.

Kind regards,

Wagdy M. Eldehna, Ph.d

Academic Editor

PLOS ONE

Journal Requirements:

2. Thank you for stating the following financial disclosure: “This work was supported by the Heilongjiang Postdoctoral Fund（Grant LBH-Z23036），2021 Central Heilongjiang University of Chinese Medicine funded projects（15041240014），Government's Plan to Support the Talent Training Project of the Reform and Development Fund of Local Universities, Heilongjiang Provincial Natural Science Foundation of China (Grant YQ2023H024)”

3. We note that your Data Availability Statement is currently as follows: “All relevant data are within the manuscript and in Supporting Information files.”

Please confirm at this time whether or not your submission contains all raw data required to replicate the results of your study. Authors must share the “minimal data set” for their submission. PLOS defines the minimal data set to consist of the data required to replicate all study findings reported in the article, as well as related metadata and methods (https://journals.plos.org/plosone/s/data-availability#loc-minimal-data-set-definition). For example, authors should submit the following data: - The values behind the means, standard deviations and other measures reported; - The values used to build graphs; - The points extracted from images for analysis. Authors do not need to submit their entire data set if only a portion of the data was used in the reported study. If your submission does not contain these data, please either upload them as Supporting Information files or deposit them to a stable, public repository and provide us with the relevant URLs, DOIs, or accession numbers. For a list of recommended repositories, please see https://journals.plos.org/plosone/s/recommended-repositories. If there are ethical or legal restrictions on sharing a de-identified data set, please explain them in detail (e.g., data contain potentially sensitive information, data are owned by a third-party organization, etc.) and who has imposed them (e.g., an ethics committee). Please also provide contact information for a data access committee, ethics committee, or other institutional body to which data requests may be sent. If data are owned by a third party, please indicate how others may request data access.

6. PLOS ONE now requires that authors provide the original uncropped and unadjusted images underlying all blot or gel results reported in a submission’s figures or Supporting Information files. This policy and the journal’s other requirements for blot/gel reporting and figure preparation are described in detail at https://journals.plos.org/plosone/s/figures#loc-blot-and-gel-reporting-requirements and https://journals.plos.org/plosone/s/figures#loc-preparing-figures-from-image-files. When you submit your revised manuscript, please ensure that your figures adhere fully to these guidelines and provide the original underlying images for all blot or gel data reported in your submission. See the following link for instructions on providing the original image data: https://journals.plos.org/plosone/s/figures#loc-original-images-for-blots-and-gels. In your cover letter, please note whether your blot/gel image data are in Supporting Information or posted at a public data repository, provide the repository URL if relevant, and provide specific details as to which raw blot/gel images, if any, are not available. Email us at plosone@plos.org if you have any questions.

Reviewers' comments:

Reviewer's Responses to Questions

**Comments to the Author**

1. Is the manuscript technically sound, and do the data support the conclusions?

Reviewer #1: Yes

Reviewer #2: Yes

2. Has the statistical analysis been performed appropriately and rigorously? 

Reviewer #1: Yes

Reviewer #2: Yes

3. Have the authors made all data underlying the findings in their manuscript fully available?

Reviewer #1: Yes

Reviewer #2: Yes

4. Is the manuscript presented in an intelligible fashion and written in standard English?

Reviewer #1: Yes

Reviewer #2: Yes

5. Review Comments to the Author

Reviewer #1: The study provide nice insights for using the natural products in tumor suppression. The manuscript can be improved more. Here are some points:

1. The authors reported the effect on BCL2, how about MCL1 and BCLxl

2. ( critical) the authors showed show the effect of the drug on non-cancerous cells or fibroblasts relevant to hepatocytes

3. I was wondering if the drug induce an autophagic or senescence response ?

4. What is meaning of media contain 1640?

5. The paper need more organization and collecting the subfigures into a main panel for each section, not scattered figures.

6. Are the authors able to perform an in vivo/3D culture study?

Reviewer #2: I have gone through the submitted manuscript titled, Inotodiol induces hepatocellular carcinoma apoptosis by activation of MAPK/ERK pathway. This study emphasizes the antitumor effect of Inotodiol against hepatocellular carcinoma illustrating its possible mechanistic effect. The manuscript is well-written and the claims are supported by the data. This manuscript will benefit from some minor revisions.

1. Little grammatical mistakes including spaces should be revised.

2. The abbreviation section should be added.

3. Funding sources should be illustrated in the manuscript.

4. Reference no 5 should be written in English.

5. In the material and method section, the reference for the migration rate formula in wound healing assay should be mentioned.

6. In figure 6 (A&B) and (F&G), it is preferred to put the statistical analysis of the western blot beside the blot figure in the same figure

6. PLOS authors have the option to publish the peer review history of their article (what does this mean?). If published, this will include your full peer review and any attached files.

Reviewer #1: **Yes: **Ahmed M. Elshazly

Reviewer #2: No

---

## [Author Response · Author response to Decision Letter 0]

25 Nov 2024

Nov 24, 2024

Dear Dr. Wagdy M. Eldehna, 

Thank you very much for your e-mail dated 11-Oct-2024.

Now I am sending our revised manuscript entitled “Inotodiol induces hepatocellular carcinoma apoptosis by activation of MAPK/ERK pathway” (Manuscript number: PONE-D-24-40336) for which we received your e-mail and reviewers’ comments 11-Oct-2024.

We have carefully revised our manuscript according the reviewer's comments. Please find a revised manuscript marked in red showing major changes and the response to the reviewer's comments.

We hope this revised manuscript is acceptable. 

With best regards,

Yours Sincerely

Yushuang Xing，

Heilongjiang University of Traditional Chinese Medicine

Harbin 150040, China. 

e-mail: 342937606@qq.com

The responses to the Reviewer's comments:

Reviewer 1:

Comments to the Author

The study provide nice insights for using the natural products in tumor suppression. The manuscript can be improved more. Here are some points：

Response: Thank you very much for your positive comments.

1.The authors reported the effect on BCL2, how about MCL1 and BCLxl

Response: We have carefully reviewed your valuable suggestions regarding the supplementation of experimental data. Following your advice, we have conducted additional experiments to investigate the expression of Bcl-XL and MCL1 in this study. We employed Western blot to detect the expression levels of Bcl-XL and MCL1 proteins, with the specific results presented in Fig 5C and Fig 5D. 

2. (critical) the authors showed show the effect of the drug on non-cancerous cells or fibroblasts relevant to hepatocytes

Response: In accordance with the suggestions you previously offered, we have supplemented the CCK8 experimental results regarding the effects of inotodiol on human liver cell line THLE-2 and mouse liver cell line AML12, with the specific results shown in Fig 1A-B.

3.I was wondering if the drug induce an autophagic or senescence response?

Response: Regarding senescence and autophagy, these are also issues that we are very interested in. We have supplemented experiments on the impact of inotodiol on the senescence and autophagy phenotypes of HepG2 and sk-hep-1 cells. The results of these experiments have been organized and presented in the supplementary data, presented in Fig S1 and Fig S2.

4. What is meaning of media contain 1640?

Response: These section have been revised in the manuscript based on your excellent suggestions.

5. The paper need more organization and collecting the subfigures into a main panel for each section, not scattered figures.

Response: Thank you for your feedback, we have adjusted the figures and presented them in a single integrated figure for the same panel.

6. Are the authors able to perform an in vivo/3D culture study?

Response: In response to your suggestions, we have supplemented our study with in vivo experiments to further validate the inhibitory activity of inotodiol against liver cancer. The specific experimental methods have been added to the manuscript and marked in red for easy identification. The results of the supplementary experiments are detailed in Fig 7A-B and Table 1. 

Reviewer 2:

Comments to the Author

I have gone through the submitted manuscript titled, Inotodiol induces hepatocellular carcinoma apoptosis by activation of MAPK/ERK pathway. This study emphasizes the antitumor effect of Inotodiol against hepatocellular carcinoma illustrating its possible mechanistic effect. The manuscript is well-written and the claims are supported by the data. This manuscript will benefit from some minor revisions.

Response: Thank you very much for your positive comments.

1.Little grammatical mistakes including spaces should be revised.

Response: Thanks for your suggestion. We tried our best to improve the manuscript and made some changes to the manuscript. These changes will not influence the content and framework of the paper. And here we did not list the changes but marked in red in the revised paper. We appreciate for Reviewers’ warm work earnestly and hope that the correction will meet with approval.

2.The abbreviation section should be added.

Response: In response to your valuable feedback, we have added a glossary of abbreviations at the end of the article to facilitate readers in consulting the professional terms and abbreviations used in the article more conveniently. This section lists in detail all the abbreviations and their full names that appear in the article, which helps to improve the readability and understandability of the article.We greatly appreciate your suggestion and believe that this improvement will make the article more complete. 

3.Funding sources should be illustrated in the manuscript.

Response: Following your suggestion, we have added a section on funding sources in the manuscript to ensure transparency and comply with academic publishing standards. This section provides detailed information on the funding sources that supported this research, including the names of the funding agencies, project numbers, and their specific contributions to the study.

4.Reference no 5 should be written in English.

Response: Following your valuable feedback, in order to enhance the international scope of the article and facilitate retrieval and comprehension for international readers, we have replaced reference 5 with the corresponding English-language literature. We have ensured that the content of the cited English literature is consistent with the research results and conclusions of the original reference 5, to maintain the academic rigor and accuracy of the article.

5.In the material and method section, the reference for the migration rate formula in wound healing assay should be mentioned.

Response: Following your suggestion, we have added reference 13 to further support the migration rate formula we proposed in the wound healing assay. This reference provides methods for calculating the migration rate in wound healing assays, which aligns with our research methods, thereby enhancing the credibility and scientific rigor of our study results.

6.In figure 6 (A&B) and (F&G), it is preferred to put the statistical analysis of the western blot beside the blot figure in the same figure

Response: Thank you for your suggestion, we have integrated the results of the same panel onto a single figure for ease of review.

---

## [Decision Letter · Decision Letter 1]

11 Dec 2024

PONE-D-24-40336R1Inotodiol induces hepatocellular carcinoma apoptosis by activation of MAPK/ERK pathwayPLOS ONE

Dear Dr. Xing,

Thank you for submitting your manuscript to PLOS ONE. After careful consideration, we feel that it has merit but does not fully meet PLOS ONE’s publication criteria as it currently stands. Therefore, we invite you to submit a revised version of the manuscript that addresses the points raised during the review process.

We look forward to receiving your revised manuscript.

Kind regards,

Wagdy M. Eldehna, Ph.d

Academic Editor

PLOS ONE

Journal Requirements:

Additional Editor Comments:

The resolution of the figures is very low, in addition, the organization of the figures and the paper is still poor. Please address these issues.

Please address the points raised by Reviewer #1 in your revision.

Reviewers' comments:

Reviewer's Responses to Questions

**Comments to the Author**

1. If the authors have adequately addressed your comments raised in a previous round of review and you feel that this manuscript is now acceptable for publication, you may indicate that here to bypass the “Comments to the Author” section, enter your conflict of interest statement in the “Confidential to Editor” section, and submit your "Accept" recommendation.

Reviewer #1: All comments have been addressed

Reviewer #2: All comments have been addressed

2. Is the manuscript technically sound, and do the data support the conclusions?

Reviewer #1: Yes

Reviewer #2: Yes

3. Has the statistical analysis been performed appropriately and rigorously? 

Reviewer #1: Yes

Reviewer #2: Yes

4. Have the authors made all data underlying the findings in their manuscript fully available?

Reviewer #1: Yes

Reviewer #2: Yes

5. Is the manuscript presented in an intelligible fashion and written in standard English?

Reviewer #1: Yes

Reviewer #2: Yes

6. Review Comments to the Author

Reviewer #1: Thanks for the authors to address all the comments. The last thing I recommend is that authors should mention the toxic effect of this compound in the discussion since its actually toxic to the normal hepatic cells. Furthermore, the authors should also report if any of the animals showed signs of toxicity.

Reviewer #2: (No Response)

7. PLOS authors have the option to publish the peer review history of their article (what does this mean?). If published, this will include your full peer review and any attached files.

Reviewer #1: **Yes: **Ahmed M. Elshazly

Reviewer #2: No

---

## [Author Response · Author response to Decision Letter 1]

9 Jan 2025

Jan 9th, 2025

Dear Dr. Wagdy M. Eldehna, 

Thank you very much for your e-mail dated 12-Dec-2024.

Now I am sending our revised manuscript entitled “Inotodiol induces hepatocellular carcinoma apoptosis by activation of MAPK/ERK pathway” (Manuscript number: PONE-D-24-40336R1) for which we received your e-mail and reviewers’ comments 12-Dec-2024.

We have carefully revised our manuscript according the reviewer's comments. Please find a revised manuscript marked in blue showing major changes and the response to the reviewer's comments.

We hope this revised manuscript is acceptable. 

With best regards,

Yours Sincerely

Yushuang Xing，

Heilongjiang University of Traditional Chinese Medicine

Harbin 150040, China. 

e-mail: 342937606@qq.com

The responses to the Editor's comments:

Journal Requirements:

Response: Thank you for your feedback and suggestions. In response to the journal's requirements, we have conducted a thorough review of all the references cited in our manuscript. We have replaced two references and have also added the DOI numbers for four articles. The revised sections are indicated in blue within the revised manuscript for easy identification.

Additional Editor Comments:

The resolution of the figures is very low, in addition, the organization of the figures and the paper is still poor. Please address these issues.

Response: Thank you for your guidance. As requested, we have made the necessary adjustments to the figures in our manuscript. We have reorganized some of the images and increased their resolution to ensure that they meet the journal's standards for clarity and quality.

Richard Ente Claros:

Please provide the unadjusted and uncropped images underlying all blot and gel figures at this time.

Response: Thank you for your reminder. We have already uploaded the unadjusted and uncropped images to the Supporting Information and named them as required.

The responses to the Reviewer's comments:

Reviewer 1:

Comments to the Author

Thanks for the authors to address all the comments. The last thing I recommend is that authors should mention the toxic effect of this compound in the discussion since its actually toxic to the normal hepatic cells. Furthermore, the authors should also report if any of the animals showed signs of toxicity.

Response: Thank you for your valuable suggestions. We have added the effects of inotodiol on the body weight and organ mass of mice in the results section, and the results showed no significant differences. At the same time, we have expanded the discussion to include the toxicity of inotodiol's in vivo and in vitro effects, with the revised content highlighted in blue in the revised manuscript.

---

## [Editor Report · Decision Letter 2]

16 Jan 2025

Inotodiol induces hepatocellular carcinoma apoptosis by activation of MAPK/ERK pathway

PONE-D-24-40336R2

Dear Dr. Xing,

We’re pleased to inform you that your manuscript has been judged scientifically suitable for publication and will be formally accepted for publication once it meets all outstanding technical requirements.

Kind regards,

Wagdy M. Eldehna, Ph.d

Academic Editor

PLOS ONE
---

## [Editor Report · Acceptance letter]

19 Jan 2025

PONE-D-24-40336R2 

PLOS ONE

Dear Dr. Xing, 

I'm pleased to inform you that your manuscript has been deemed suitable for publication in PLOS ONE. Congratulations! Your manuscript is now being handed over to our production team.

Kind regards, 

on behalf of

Dr. Wagdy M. Eldehna 

Academic Editor

PLOS ONE